# Toward a Unifying Hypothesis for Redesigned Lipid Catabolism as a Clinical Target in Advanced, Treatment-Resistant Carcinomas

**DOI:** 10.3390/ijms241814365

**Published:** 2023-09-21

**Authors:** Paul M. Bingham, Zuzana Zachar

**Affiliations:** Biochemistry and Cell Biology, Stony Brook University, Stony Brook, NY 11794, USA; zuzana.zachar@stonybrook.edu

**Keywords:** cancer, metabolism, resistance, lipid, fatty acid, CPI-613, thioridazine, crizotinib

## Abstract

We review extensive progress from the cancer metabolism community in understanding the specific properties of lipid metabolism as it is redesigned in advanced carcinomas. This redesigned lipid metabolism allows affected carcinomas to make enhanced catabolic use of lipids in ways that are regulated by oxygen availability and is implicated as a primary source of resistance to diverse treatment approaches. This oxygen control permits lipid catabolism to be an effective energy/reducing potential source under the relatively hypoxic conditions of the carcinoma microenvironment and to do so without intolerable redox side effects. The resulting robust access to energy and reduced potential apparently allow carcinoma cells to better survive and recover from therapeutic trauma. We surveyed the essential features of this advanced carcinoma-specific lipid catabolism in the context of treatment resistance and explored a provisional unifying hypothesis. This hypothesis is robustly supported by substantial preclinical and clinical evidence. This approach identifies plausible routes to the clinical targeting of many or most sources of carcinoma treatment resistance, including the application of existing FDA-approved agents.

## 1. Introduction

Successful clinical targeting of carcinoma metabolism requires that we develop a relatively complete, coherent, predictive theory of how normal cell and carcinoma cell metabolism differ. Our goal here is narrow: to review the current status of an important element of this effort. We will endeavor to contextualize one specific area of progress: changes in the properties and regulation of lipid catabolism that are strongly associated with and apparently cause treatment resistance in advanced carcinomas.

We organize this discussion around an emerging provisional hypothesis describing the core features of this process. Sarcomas and liquid tumors can show some of the lipid metabolic features we focus on here, as we will briefly mention below. However, the story we review is largely based on the much more extensive body of data from carcinomas; we will largely focus throughout on this tumor class.

It is useful to begin with the longstanding insight that carcinomas are a particular class of deranged tissue. Specifically, carcinomas pervasively resemble healing wounds that never resolve to complete the normal healing process [1,2,3,4,5,6,7,8]. These insights strongly imply that features of carcinoma metabolism will reflect the mobilization and strategic modification of evolved mechanisms originally designed to support wound healing, as follows:

First is the reregulation of catabolism to deal with highly fluctuating and often inadequate access to nutrients and oxygen. Among these challenges is supporting catabolism sufficient for survival and ultimate growth without the generation of toxic levels of reactive oxygen species (ROS) [8,9,10,11].

The second is the reregulation of cell death decision-making. Cells in normal healing wounds need to make an adaptive choice between programmed cell death (commonly colloquially referred to as the “scab” response in wound healing literature) or survival, with retention of the capacity for growth and/or differentiation (the “scar” response). We expect the genetic and epigenetic evolution of carcinomas to mobilize (and refine) catabolic processes in support of growth but also to avoid the cell death responses that normal healing wound cells would undergo. Effective therapeutic assault must be sufficient to overcome this cell death resistance.

The key implications of this broader picture are as follows: Lipids are an especially important nutrient for stressed carcinoma cells and normal healing wound epithelial cells. This unique value of lipids results from their exceptional density and economy of storage of energy/reducing potential, and the crucial role of lipids as structural molecules in membranes and, occasionally, as signals [12,13,14]. The downside of lipids in the carcinoma/wound environment is that their exploitation for energy is very strongly dependent on oxygen, again, a resource commonly severely limited and variable in this context.

Thus, we expect the evolved wound healing machinery to be constructed to allow individual epithelial cells to control flux through lipid metabolism based on oxygen availability, both to support survival/growth behavior when oxygen is adequate and to trigger cell death when oxygen levels are too low to match lipid availability. In turn, as noted, we expect this machinery to be coopted and reprogrammed in the evolution of advanced malignancy, including attenuation of the cell death response.

A large body of work on lipid metabolism in advanced, treatment-resistant carcinomas fits well with this picture (reviewed below). Moreover, because the founding machinery is relatively exotic (wound healing), its modified version, driving broad treatment resistance in advanced carcinomas, is expected to have features that can be targeted in patients without unacceptable side effects.

The following section is a high-altitude overview of the provisional hypothesis we will explore.

## 2. Context of “Lipid Metabolism Resistance System” (LMRS) Hypothesis

Over the last decade, it has become increasingly clear that changes in lipid metabolism are commonly associated with advanced, treatment-resistant carcinomas [15,16,17,18,19,20,21,22,23,24,25,26,27,28]. There is extensive evidence for ongoing redesign of lipid metabolism in tumor cells as they progress to more advanced, treatment-resistant carcinoma [29,30,31,32,33,34,35,36]. Moreover, hypoxia, commonly experienced by carcinoma cells in vivo and representing a negative clinical prognostic, upregulates fatty acid uptake and storage [37,38,39]. Hypoxia interacts with ROS production in various ways beyond those we will discuss in detail below. These interactions include those mediated by known regulators, such as HIF1 and others [40,41,42,43,44,45].

Finally, clonal selection processes operate on carcinoma stem cells in tumors [46]. These processes generate cell subpopulations wherein lipid storage/catabolism upregulation and ensuing treatment resistance become characteristic of the advanced carcinomas they found.

While other features of carcinoma metabolism might also be of therapeutic interest [47,48,49,50], our primary focus here is the putatively universal core features of lipid catabolism specifically involved in producing treatment-resistant carcinomas (Figure 1).

We will refer to this provisional picture as the “lipid metabolism resistance system” or LMRS (Figure 1). One of the central assumptions of this hypothesis is that elevated lipid catabolism provides increased access to energy and reduces potential. These increases, in turn, are proposed to allow carcinoma cells to better survive and recover from diverse therapeutic assaults, thereby rendering them broadly treatment-resistant.

The precise biochemical or cellular role(s) of LMRS metabolism in supporting carcinoma cell survival, growth, and treatment resistance remains to be clearly characterized. At present, experimental data are sufficient to support this causal relationship but inadequate to yield a fully detailed mechanistic picture of the basis of this effect.

We review in detail below the steps in this specialized, resistance-associated catabolism of fatty acids in carcinoma cells. This redesign for elevated use and dependence upon lipid catabolism begins with generating the lipid substrates supporting this process. This frequently includes the elevated capacity of carcinoma cells to synthesize fatty acids de novo [16,51,52,53]. Also common is the enhanced carcinoma cell ability to actively import fatty acids and their multimeric composites from their surroundings [26,54,55,56,57,58]. Finally, carcinoma cells interact with and reciprocally manipulate surrounding stromal cells to procure elevated access to the lipids these support cells can manufacture and export [32,57,59,60,61,62,63,64,65,66,67,68,69,70,71,72,73,74,75,76].

In spite of their importance, these diverse processes entail sufficient redundancy that attempting to clinically target individual components is likely to be ineffective and highly susceptible to evolved resistance. Thus, we will not emphasize elevated carcinoma cell lipid access further here. Rather, we will focus on the catabolic processing of fatty acids once they arrive in advanced, treatment-resistant carcinoma cells.

## 3. Detailed Description of the Steps in the LMRS Hypothesis

The LMRS process includes three major steps (Figure 1). First is peroxisomal fatty acid beta-oxidation, initiated by acyl-coenzyme A oxidases (Acox). Second, is desaturation, resulting in a monounsaturated fatty acid (MUFA) derived from saturated fatty acid precursors and commonly catalyzed by stearoyl-CoA desaturase (SCD1). This step may precede or follow Acox oxidation. Finally, the products of these first two steps are transferred to the mitochondrion for complete oxidation to CO_2_ and H_2_O.

It is crucial to recognize that all these steps are oxygen-dependent in ways that allow catabolic flux of fatty acids to be controlled on the basis of real-time oxygen availability.

We review each of these steps and its relevance to carcinoma treatment resistance in detail as below.

### 3.1. Mitochondrial Oxygen-Dependent Metabolism of Preprocessed Fatty Acids

The LMRS hypothesis is more easily grasped by beginning at the terminal mitochondrial steps. As synopsized above, the presumptive role of the preceding, extra-mitochondrial steps in fatty acid catabolism is to control flux in response to limited and potentially fluctuating oxygen availability (in wound healing or carcinoma microenvironments). Both peroxisomal beta-oxidation and desaturase processing require molecular oxygen, rendering these steps “licensed” and rate-controlled by oxygen availability (Section 3.2 and Section 3.3).

To grasp our provisional interpretation of this process, it is essential to understand how the final mitochondrial steps of fatty acid catabolism are, likewise, managed in response to low, variable oxygen levels. Oxygen regulation of LMRS fatty acid catabolism is pervasive from the beginning to the end of the process.

When oxygen availability is limited, several mitochondrial processes generate elevated levels of reactive oxygen species, ultimately or directly in the form of hydrogen peroxide, H_2_O_2_, as discussed in more detail below [77]. We note that other ROS species are produced in some of these processes, especially superoxide; however, these other ROS species are generally either rapidly destroyed or converted to H_2_O_2_. Moreover, H_2_O_2_ appears to be the primary informational ROS molecule in the cases we discuss [77]. Thus, it is reasonable to focus on hydrogen peroxide.

On the one hand, the mitochondrial electron transport complexes (ETC) retain elevated steady-state levels of electrons when these cannot be off-loaded at the end step in the ETC system (complex IV; cytochrome c oxidase) to form water from oxygen. Instead, these excess electrons are donated directly by intermediate ETC steps to whatever oxygen remains, creating H_2_O_2_ [77]. On the other hand, matrix oxoacid dehydrogenase complexes, especially alpha-ketoglutarate dehydrogenase (KGDH), also donate reducing potential to residual oxygen (again, making H_2_O_2_) when their NADH product levels become elevated in the absence of the ability for NADH electrons to be efficiently passed off to a now-saturated ETC [78,79,80,81].

Mitochondrial metabolism is expected to have evolved to control this hypoxia-driven elevated ROS generation. Such control appears to be reflected in the adaptive redox regulation of several steps through H_2_O_2_-dependent modification. This redox blockade has the overall effect of suppressing further fatty acid flux in the face of oxygen deprivation-dependent clogging of mitochondrial metabolism (and, thus, increased H_2_O_2_ production).

On the one hand, elevated mitochondrial matrix H_2_O_2_ levels suppress further fatty acid import. This includes redox modification of the matrix face of the CAC transporter (mitochondrial carnitine acyl-carnitine carrier, Figure 1; SLC25A20), which blocks fatty acyl coenzyme A thioester import across the inner mitochondrial membrane [82].

On the other hand, several rate-limiting steps in the TCA cycle oxidation of fatty acid-derived acetate units are, likewise, highly sensitive to redox inhibition in the presence of elevated H_2_O_2_ levels [83]. These include aconitase [84] and KGDH [78,79,80,81,85] (Figure 1). Notably, H_2_O_2_ produced by KGDH constitutes a local negative feedback loop controlling this enzyme itself, in addition to contributing more broadly to the determination of regulatory levels of matrix redox signals [78,79]. These redox inhibition steps choke off further fatty acid-dependent electron generation when oxygen levels are insufficient.

As discussed in Castelli et al. [42,86,87] and Ying and Hu [88], mitochondrial fatty acid oxidation (FAO) can generate elevated levels of reactive oxygen species (ROS), while, reciprocally, driving antioxidant control of ROS. The relative rates of these two opposing processes are strongly influenced by oxygen availability, as described below.

There are various mitochondrial enzymatic systems for generating the antioxidant capacity needed to manage/eliminate metabolism-generated ROS. Among the most important of these is the transhydrogenase complex (NNT) [81,89,90,91,92]. The NNT complex spans the inner mitochondrial membrane and exploits energy stored in the Mitchell/Moyle proton gradient to drive the transfer of reducing potential from NADH to NADPH. Major mitochondrial antioxidant systems commonly require NADPH as a direct or ultimate source of reducing potential [77]. When oxygen levels are limited, one of the drivers of the NNT forward reaction, the proton gradient, will be attenuated as a result of the clogged electron transport through the inner membrane ETC hydrogen ion pumps. This effect, in turn, depresses levels of NADP+ reduction to NADPH and, thus, further enhances steady-state H_2_O_2_ levels (again, in turn, lowering catabolic flux through the TCA cycle as discussed above).

NNT control of mitochondrial matrix redox levels is especially important in LMRS catabolism. Specifically, NNT is potently inhibited by saturated fatty acid CoA thioesters, the primary matrix form of these fatty acids [93,94,95]. Thus, if levels of saturated fatty acids (especially palmitate and stearate) are elevated in the matrix as a result of desaturation failure in the absence of adequate cytosolic oxygen levels (Section 3.3), mitochondrial matrix H_2_O_2_ levels are expected to show an additional increase, suppressing further fatty acid import and catabolism (above).

In view of these roles in managing oxygen-limited mitochondrial metabolism, it is plausible that the NNT system will be reregulated in advanced carcinomas, as indicated below.

Illuminating is the recent work of Han et al., who showed that elevated NNT enzymatic activity enhanced gastric carcinoma mitochondrial metabolism and resistance to immunotherapy [96]. Moreover, an upstream trigger for the post-translational modification producing this elevated NNT activity is IL-1beta. This trigger, in turn, is sufficient to drive local inflammation and, ultimately, gastric carcinoma. Thus, it is reasonable to anticipate that this inflammation/carcinoma-dependent activation of NNT might be broadly general.

Further supporting this picture, genetic ablation of NNT interferes with malignancy in preclinical systems, including adrenal carcinoma [97], renal carcinoma [98], gastric carcinoma [99], and nonsmall cell lung carcinoma (NSCLC) [100].

Notably, KGDH and NNT appear to interact intimately (perhaps directly), as evidenced by their coparticipation in reductive mitochondrial citrate synthesis [101]. Moreover, there is circumstantial evidence that KGDH redox regulatory processes are modified in carcinoma cells [78].

Finally, NNT reregulation in carcinomas occasionally involves downregulation rather than the more common upregulation discussed above [102,103]. It remains to be seen what the implications of such minority observations are for the generality of the LMRS hypothesis in its simplest form.

### 3.2. Acox-Catalyzed, Oxygen-Dependent Peroxisomal Fatty Acid Beta-Oxidation

There is substantial evidence that peroxisomes interact with mitochondria, including through direct contact, and thereby contribute to the control of mitochondrial lipid metabolism [104,105,106,107,108]. Moreover, these peroxisomal/mitochondrial interactions are directly implicated in determining carcinoma lipid metabolism [109,110]. Especially notable here, elevated expression of mitochondrial/peroxisomal interaction mediators has negative clinical prognostic implications for both sarcomas and carcinomas [111].

Sarcomas yield a particularly useful insight. Elevation of one of these peroxisomal/mitochondrial interaction regulators (phosphatidylinositol-5-phosphate 4-kinase type 2 alpha; PI5P4Kalpha) is also correlated with advanced, aggressive status in sarcomas (op cit.). Moreover, PI5P4Kalpha knockout dramatically reduced KRAS/p53KO-dependent sarcoma formation in a mouse system and the growth of in vivo tumors from cell lines derived from these primary tumors (op cit.).

The beta-oxidation of fatty acids in peroxisomes is of special interest for clinical targeting. Specifically, peroxisomal beta-oxidation is very commonly essential for advanced, treatment-resistant carcinoma cells (below). In contrast, differentiated normal cells appear to rely largely or entirely on direct mitochondria-autonomous uptake systems to support fatty acid oxidation [112].

Unlike the analogous mitochondrial process, peroxisomal fatty acid beta-oxidation is initiated by an oxidase reaction that directly burns an oxygen molecule. This process transfers the abstracted electrons to oxygen with the production of hydrogen peroxide. This H_2_O_2_ is then rapidly destroyed by catalase in the peroxisomal matrix. This initial fatty acid reaction is catalyzed by the Acox oxidases. This process has the effect of licensing the ultimate transfer of shortened fatty acids and acetate units to the mitochondrion on the basis of the presence of adequate levels of molecular oxygen to carry out the oxidase reaction [109,113].

Several studies robustly support a central role for peroxisomal beta-oxidation of fatty acids in advanced, treatment-resistant carcinomas, as predicted by the LMRS hypothesis.

First, Cai et al. showed that peroxisomes were essential to liver carcinoma cell survival in vitro and xenograft growth in vivo [114]. Remarkably, the same gene knockout/knockdown (PEX2) that eliminates peroxisome function in these carcinoma cell studies nonetheless permits normal embryonic development. This observation directly supports the other evidence discussed herein that the pattern of peroxisomal metabolism seen in carcinomas is preferentially important to these cancers.

Second, a later step in the peroxisomal fatty acid beta-oxidation pathway generates reduced NADH. Unlike the FADH_2_ intermediate generated by the first Acox step, these electrons cannot be transferred to molecular oxygen for H_2_O_2_ production. Instead, NAD+ must be regenerated through mitochondrial shuttles, most importantly a lactate/pyruvate shuttle. This shuttle appears to depend on the monocarboxylate transporter isoform, MCT2. MCT2 is a high-affinity transporter of pyruvate and lactate across membrane bilayers. Unlike other members of this transporter family that are localized to the plasma membrane, MCT2 commonly localizes to the peroxisomal membrane (and probably also the mitochondrial membrane) [115]. Crucially, in this context, Valenca et al. have shown that MCT2 is upregulated in prostate carcinoma and that peroxisomal MCT2 localization is essential to prostate carcinoma cell growth [116,117].

Of particular relevance to the LMRS hypothesis, this MCT2 shuttle is limited by oxygen-dependent mitochondrial disposal of electrons from lactate conversion back to pyruvate [118]. Thus, this peroxisomal shuttle shares the feature of oxygen availability-licensing of lipid catabolism with the Acox oxidase and desaturase reactions (Section 3.3 below).

Third, BRAF mutant melanoma cells generate cell subclones, dubbed persisters, that are resistant to clinical drug targeting of this kinase. These persister cells are thought to be the major source of clinical drug resistance. Shen et al. showed that these drug-resistant cells are dependent on peroxisomal fatty acid oxidation [119]. Genetic knockdown or drug inhibition of the peroxisomal Acox1 oxidase eliminates these persister cells.

Fourth, the carnitine O-octanoyltransferase (CROT) and carnitine acetyltransferase (CRAT) transporters are responsible for moving shortened fatty acid and acetate units from peroxisomal fatty acid beta-oxidation to the mitochondrion for completion of the oxidation process. Lasheras-Otero et al. showed that these transporters are essential for the survival of detached melanoma cells in vitro (proxies for circulating carcinoma cells in vivo) and that suppression of CROT or CRAT substantially reduces melanoma metastasis in in vivo preclinical models [120]. Moreover, these effects of CROT/CRAT knockdown can be mimicked with a small-molecule Acox oxidase inhibitor (op cit.).

Fifth, our group characterized carcinoma resistance to the CPI-613 tumor-specific TCA cycle inhibitor [121]. We showed that in vitro resistance to this drug depended on elevated accumulation and catabolism of lipid stores. We were able to block this resistance with the small-molecule Acox inhibitor, thioridazine. Moreover, thioridazine was also potent in sensitizing in vivo xenografts of an otherwise CPI-613-resistant pancreatic carcinoma (PDAC) cell line (AsPC1; op cit.). In vitro studies indicated that CPI-613 resistance was dependent on peroxisomal beta-oxidation in every carcinoma cell type tested. Further, inhibitor studies implicate acute MET kinase signaling in maintaining this LMRS lipid-dependent peroxisomal rescue pathway. The FDA-approved MET inhibitor (crizotinib; CRZ) sensitized resistant AsPC1 PDAC xenografts analogously to thioridazine (op cit.).

A substantial body of evidence further implies an important role of peroxisomal fatty acid catabolism in carcinoma treatment resistance. In spite of the reduction in total amounts/numbers of peroxisomes in several carcinoma cases [122,123], there is also a common observation of an elevation in specific peroxisomal enzymatic activities in various cancers, including carcinomas.

Dahabieh et al. showed that the upregulation of peroxisomal enzymes supports lymphoma cell resistance to an HDAC inhibitor anticancer agent [124]. Further, Dahabieh et al. showed that reducing peroxisome levels through stimulation of plexophagy sensitized resistant lymphoma cells to this HDAC inhibitor [125]. Zheng et al. showed that Acox1 levels determined doxorubicin sensitivity in lymphoma cells [126].

Kim et al. found that strong Acox1 expression was a significant negative clinical prognostic for breast carcinoma [127]. Yu et al. showed that Acox activity supports prostate carcinoma cell proliferation, migration, and invasion in vitro [128]. Kuna et al. showed that peroxisomal beta-oxidation can be a robust source of fatty acid catabolism in an NSCLC carcinoma cell line [129].

Tamatani et al. [130] and Okamoto et al. [131] showed that Acox1 overexpression can exhibit oncogene-like activity, permitting in vivo xenograft carcinoma formation by otherwise nonmalignant cells. These striking results indicate that the elevation of LMRS-like lipid metabolism is sufficient to robustly reduce barriers to carcinoma formation.

Finally, genetic ablation of peroxisomes in Drosophila produces lethality at the pupal stage. Larval development happens normally, irrespective of peroxisome knockout. Pupal development involves massive histolysis of larval tissues, followed by extensive stem cell division. It is during these processes that the peroxisomal knockout produces lethality. Thus, pupal development likely makes use of wound-healing machinery, and this process is significantly and selectively compromised without peroxisomes [132].

These authors made a further striking observation. They showed that these consequences of peroxisome loss in Drosophila can be largely eliminated by providing a mixture of saturated and unsaturated medium-chain-length fatty acids in the diet. In context, these results indicate that the pupal developmental requirement for peroxisomes can be replaced by the products of peroxisomal beta-oxidation, the shortening of fatty acid chain length. Further consistent with this view, the loss of mammalian melanoma stem cell populations through peroxisomal attenuation can be rescued by comparable medium-chain-length fatty acid supplementation [120]. In aggregate, these data support the picture that beta-oxidation-dependent provision of shortened fatty acids is the peroxisomal process sustaining clinically relevant carcinoma cell behavior, consistent with the proposed details of the LMRS hypothesis (Figure 1).

### 3.3. Oxygen-Dependent Fatty Acid Desaturation by the SCD1 Enzyme Anchored on the Cytoplasmic Face of the Endoplamic Reticulum (ER)

Stearoyl-coenzyme A desaturase 1 (SCD1) is the most thoroughly characterized component of LMRS (Figure 1), providing the most robust and extensive opportunities to test this hypothesis.

Key evidence indicates that SCD1 participates in normal tissue wound healing, as the LMRS hypothesis predicts [133,134]. Likewise, the well-defined mouse adipocyte-supported hair follicle regeneration/wound healing system is dependent on SCD1 [135]. Also striking is the extent to which SCD1 appears to be actively involved in the control of metabolism in diverse normal cell settings [136].

SCD1 catalyzes the introduction of a double bond (desaturation) into saturated fatty acids from endogenous lipogenesis or exogenous sources. SCD1 substrates commonly include palmitate (C16:0) and stearate (C18:0), resulting in their conversion to palmitoleate (C16:1 n-7) and oleate (C18:1 n-8), respectively [137,138,139]. Among its diverse effects, this desaturation is expected to prevent saturated fatty acid inhibition of mitochondrial NNT and, thus, prevent ensuing inhibition of mitochondrial metabolism (Section 3.1).

The striking features of SCD1 function are two. First, as noted, this reaction makes direct use of oxygen and is inhibited at low oxygen levels [137,138,140]. Second, SCD1 expression is crucial to the function and survival of normal stem cells (above) and is commonly upregulated in carcinomas in general. This SCD1 upregulation is often especially extreme in carcinoma stem cells (CSCs), which are thought to be responsible for most treatment resistance. This work has been well discussed recently [141,142,143,144,145,146]. Our focus here will be on the issues most directly relevant to the provisional LMRS hypothesis.

Cases where SCD1 expression and/or upregulation are crucial in advanced malignancy, including in treatment resistance, are diverse. The following are illuminating examples: Scaglia and Igal [147] and Scaglia et al. [148] showed that SCD1 plays a crucial role in tumor cell proliferation and metabolism in transformed fibroblasts and several carcinoma cell models, including NSCLC. Xuan et al. showed that SCD1 and a related oxidase are essential to malignancy and platinum drug resistance in an ovarian carcinoma model [149]. Huang et al. found that SCD1 upregulation is associated with late-stage lung carcinoma, representing a poor clinical prognostic indicator [150]. Pisanu et al. showed that SCD1 expression is necessary for cisplatin resistance in lung carcinoma stem cells and that elevated SCD1 expression is a negative prognostic in lung carcinoma patients [151]. Pisanu et al. showed that the evolved resistance of melanomas to BRAF inhibitors depends on SCD1 [152]. Hwang et al. showed that SCD1-dependent oleic acid provision by cancer-associated fibroblasts (CAFs) was essential in efficient xenograft lung carcinoma tumor formation [153]. Fritz et al. showed that advanced prostate carcinoma clinical biopsy samples display elevated SCD1 expression and that inhibition of SCD1 robustly inhibits prostate carcinoma xenograft growth [154].

Budhu et al. [155] and Ma et al. [156] found that elevated SCD1 is a negative prognostic in hepatocellular carcinoma. Peck et al. showed that SCD1 knockdown blocked prostate carcinoma xenograft growth and found that elevated SCD1 expression levels were a negative prognostic in breast carcinoma patients [141]. von Roemeling et al. showed that SCD1 is essential to anaplastic thyroid carcinoma (ATC) cell survival and elevated SCD1 levels are a negative prognostic in ATC patients [157]. Liu et al. found that elevated SCD1 is a robustly negative prognostic in pancreatic carcinoma, thymoma, melanoma, and renal clear cell carcinoma [158]. Noto et al. found that SCD1 is a strong negative prognostic in lung carcinoma and is necessary for the preservation of the stem cell-like status correlating with drug resistance [159,160]. Ran et al. showed that SCD1 expression was required for the epithelial-mesenchyme transition associated with in vivo metastasis and found that elevated SCD1 tumor expression was a robust negative clinical prognostic in colorectal carcinoma [161].

Li et al. showed that SCD1 is essential for ovarian carcinoma cell lines to form tumors in in vivo preclinical models [162]. Bansal et al. showed that hepatocellular carcinoma cell sensitivity to various chemotherapeutic agents was predicted by their levels of SCD1 expression; these resistance effects were attenuated by suppression of SCD1 expression or activity [163]. Luo et al. showed that SCD1 expression supported radiation resistance in esophageal squamous cell carcinoma [164]. Zhang et al. showed that SCD1 was essential to stem cell-like behavior and chemoresistance in gastric carcinoma [165]. Morais et al. [166] and Parik et al. [167] showed that glioblastoma aggressiveness and temozolomide resistance depended on SCD1 expression. Lien et al. showed that in vivo PDAC growth under low lipid dietary availability was strongly dependent on SCD1 [168]. Sun et al. showed that irinotecan resistance in colorectal carcinoma cells depended on SCD1 expression [169]. Li et al. showed that SCD1 ablation substantially inhibited endometrial carcinoma xenograft tumor growth [170].

Piao et al. generated cancer stem cell derivatives of a bladder carcinoma; these CSCs show substantially more efficient xenograft formation [171]. These authors then showed that SCD1 expression was prominently upregulated in these CSCs relative to the parental cell line and that SCD1 inhibition blocked in vitro migration and invasion of these CSCs. Finally, these authors found that high SCD1 expression was a strong negative prognostic for bladder carcinoma in clinical data sets.

Wang et al. found that high SCD1 expression was a strong negative prognostic factor in renal clear cell carcinoma [172].

While these data clearly indicate the pervasive involvement of SCD1 in support of carcinoma malignancy and treatment resistance, there are rare examples of the contrary case of SCD1 downregulation in treatment resistance [173,174]. It will ultimately be important to understand the mechanistic basis of these rare opposite response patterns.

The behavior of SCD1 in carcinomas and their treatment resistance commonly conform robustly to the involvement of peroxisomal fatty acid oxidation (Section 3.2) and mitochondrial redox reregulation (Section 3.1). Collectively, these data strongly support the LMRS hypothesis for treatment resistance. These insights indicate a path forward to effectively attacking this resistance, which, in turn, apparently drives the majority of carcinoma clinical lethality (below).

## 4. Conclusions and Practical Clinical Implications

The evidence above strongly suggests that the specialized LMRS lipid catabolic pathway is crucial to sustaining many or most cases of advanced carcinoma treatment resistance. These insights should allow an attack on the especially troublesome subclasses of carcinoma cells thought to be responsible for resistance; these include cells operationally classified as carcinoma stem cells, EMT-prone carcinoma cells, and/or carcinoma tumor initiating cells.

Most importantly, available evidence suggests that targeting the LMRS pathway may sensitize carcinoma cells to most or all of the diverse agents currently in wide use for cancer therapy. These range from radiation to chemotherapy/cytotoxic treatment, non-LMRS antimetabolic agents (CPI-613, for example), and immunotherapy. This perspective indicates a specific path toward improving many clinical outcomes, as follows: Table 1 summarizes selected preclinical studies that are especially useful for clinicians in exploring options they might wish to investigate.

One FDA-approved agent, thioridazine (TZ), attacks the peroxisomal fatty acid oxidase (Acox) step in the LMRS pathway (Figure 1). TZ targets Acox as a potent off-target effect [175,176]. TZ is relatively well tolerated, and clinicians have decades of experience with its chronic use in thousands of noncancer patients [177]. Thus, TZ represents an exceptionally promising opportunity to clinically test predictions of the LMRS hypothesis. Our preclinical experience with TZ in combination with the cancer mitochondrial TCA cycle inhibitor, CPI-613, supports this picture [121]. Likewise, Shen et al. [119] and Lasheras-Otero et al. [120] used TZ in their preclinical studies, supporting the generality of TZ use in carcinomas. Should TZ prove promising in initial clinical studies, the development of more robust Acox inhibitors would be warranted and likely practical.

As noted, inhibitor studies implicate acute MET kinase signaling in maintaining the LMRS pathway [121]. Consistent with this possibility, there is substantial evidence for MET upregulation in association with advanced carcinoma treatment resistance [178,179,180,181,182,183,184,185]. It will be important to assess the generality of MET inhibitors in apparently suppressing LMRS metabolism in carcinomas. The development of new MET inhibitors thrives [186,187,188,189].

Practical clinical assessment of attacks on other steps in the LMRS process is more remote than for TZ and CRZ. However, indications of success with the clinical application of TZ and/or CRZ would create the incentives to improve the targeting of the other steps in this process. The current status of these targeting opportunities is as follows:

There is evidence that the development of clinical inhibitors selective for the MCT2 transporter, essential to sustained peroxisomal oxidation, might be practical [189,190,191,192]. Such inhibitors are expected to potentially be as clinically effective as Acox inhibitors. Likewise, inhibitors of the CROT and CRAT transporters might be effective, though we currently lack any useful test cases for these.

While the involvement of SCD1 in carcinomas and, especially, in treatment resistance seems clear, practical clinical targeting of this enzyme will require careful additional exploration. Specifically, chronic SCD1 inhibition produces undesirable side effects, largely resulting from their compromising normal stem cell function [187]. However, it will be of value to explore whether acute SCD1 inhibition on the time scale of weeks might be sufficient to contribute to robust clinical anticarcinoma effects without unacceptable side effect toxicity. Small molecule SCD1 inhibitors are known, and the development of workable clinical versions is likely realistic [193,194].

Genetic ablation of NNT is relatively innocuous in the short term in humans [92,195], indicating that targeting this activity acutely in a therapeutic context should be well tolerated. Current experimental NNT inhibitors are clinically impractical [95]. Developing small molecule agents for clinical targeting of NNT is plausible in view of the detailed NNT structural picture now available [89].

Finally, of course, additional practical barriers to targeting the LMRS process might emerge. For example, some such approaches might counterproductively suppress immune attacks in carcinomas. Aggressive clinical and in vivo preclinical investigation of LMRS-related opportunities will be important in the near future.

## Figures and Tables

**Figure 1 ijms-24-14365-f001:**
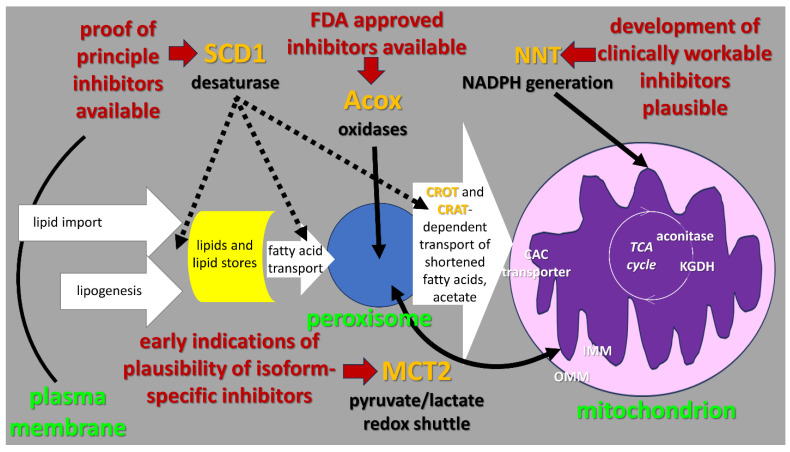
Key processes in the “lipid metabolism resistance system” (LMRS). In orange text are enzyme activities managing the specialized catabolic flow of fatty acids found in many advanced carcinomas (text). One of the central assumptions of this picture is that this system produces elevated lipid catabolism, engendering increased levels of energy and reducing potential under the conditions of in vivo carcinomas, including inadequate oxygen supplies. These increases, in turn, are proposed to allow carcinoma cells to better survive and recover from therapeutic assault, thereby rendering them broadly treatment-resistant. (IMM and OMM = inner and outer mitochondrial membranes.)

**Table 1 ijms-24-14365-t001:** In light of the large body of evidence for LMRS metabolism involvement in advanced carcinoma treatment resistance, assessing opportunities for clinical deployment of these insights is important. Several studies have been conducted that support such assessments and possible planning for future clinical trials. This table briefly summarizes the most useful of these of which we are aware.

Reference	Carcinoma	Fundamental Observation
IN VITRO (cell culture)		
[151]	lung carcinoma	MF-438 inhibition of SCD1 desaturase reduces cisplatin resistance in lung carcinoma 3D spheroids
[152]	melanoma	MF-438 inhibition of SCD1 desaturase reduces resistance to BRAFi plus MEKi inhibition of melanoma 3D spheroid formation
[167]	Glioblastoma(GBM	Merck inhibitors of SCD1 (Cpd3j) or FADS2 (SC26196) desaturases sensitize GBM cells to temozolomide-induced cell death in otherwise resistant cell lines
IN VIVO (tumor models)		
[30]	melanoma	Etomoxir fatty acid oxidation inhibitor interferes with nongenetic xenograft resistance to BRAF inhibitors (in the presence of DCA perturbation of glycolytic regulation).
[119]	melanoma	Thioridazine fatty acid oxidation inhibitors robustly block nongenetic xenograft resistance to combinations of BRAF and MEK inhibitors.
[51]	pancreatic adenocarcinoma(PDAC)	Orlistat inhibition of fatty acid synthesis robustly blocks gemcitabine resistance in orthotopic xenografts.
[121]	pancreatic adenocarcinoma(PDAC)	Thioridazine substantially interferes with xenograft resistance to tumor-specific TCA cycle inhibitor, CPI-613. Crizotinib METi mimics this in vivo thioridazine effect, apparently through LMRS interference.
[156]	hepatocellular carcinoma (HCC)	Novel inhibitor (SSI-4) suppression of SCD1 activity robustly overcomes HCC xenograft resistance to multi-RTK inhibitor, sorafinib.
[164]	esophageal squamous cell carcinoma(ESCC)	MF-438 SCD1 inhibitor moderately enhances ESCC xenograft radiation sensitivity.

## Data Availability

Not applicable.

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
