# Peer review of "Toward a Unifying Hypothesis for Redesigned Lipid Catabolism as a Clinical Target in Advanced, Treatment-Resistant Carcinomas"

_ijms, 2023, doi:10.3390/ijms241814365_

Round 1
Reviewer 1 Report
The review is interesting and well written, even if I have some concerns about it.
I apologize but a lot of sentences are redundant in their content, please the authors should avoid repetition and eliminate reduntant conceps. On the other side, important findings about hypoxia and ROS (H2O2 is not the only one reactive oxigen species important in resistance cells...) in relation to lipid metabolism/resistance are not mentioneted. Please discuss it.
Moreover, as there is a great attention for the biochemistry but not so for in vitro studies and clinical trials, I suggest to insert an esplicative table to show these.
I did not understand why authors refer to cancers/tumors in a general way. Lipid metabolism are equally important for all type of resistance carcinomas and solid cancers? What is the contribution of this lipid degerulation in drug resistant mechanism? Please, discuss them.
Clinical options discussed in the last paragraph are for all type of resistant tumors? Please, are details.
In introduction, in my opinion, the authors could avoid the use of social, sociopat, decisions and so on when they refer to cancer cells.
Author Response
Bingham and Zachar ijms-2562173
Responses to REVIEWER 1
Technical note: Revisions were done in Track Changes and line numbers below refer to the display toggled to hide revision details (standard type) or to reveal revision details (italic).
This reviewer is concerned that readers may be confused about the scope of the malignancies displaying the phenomena we review. We have edited wording at numerous points (including in the INTRODUCTION) to emphasize that our primary focus is carcinomas. It is of note that sarcomas and liquid tumors are likely to also share some of these properties; however, more work is required to decisively assess the extent of these shared features. A few sarcoma and liquid tumor cases are mentioned (and clearly labeled) in a supporting context; however, again, we focus very largely on carcinomas.
This reviewer is concerned that we do not sufficiently discuss other ROS, hypoxia interactions, and their relationships to lipid metabolism. We explain our choices to avoid certain areas of detail (for brevity and focus) in the revised lines 89-92/125-128 and 168-172/210-215. We also provide additional references to this un-discussed work in lines 89-92/125-128. We emphasize that the areas of work we do not discuss in detail do not change in any way the central conclusions and hypotheses emerging from the work we do review in detail.
This reviewer is concerned that we do not describe the detailed mechanism whereby the lipid dysregulation we review produces treatment resistance. In fact, the experimental evidence clearly, strongly supports this causal relationship and some of its regulatory features. However, this evidence is not yet adequate to define a clear, detailed mechanistic basis for this relationship. We have revised the manuscript to make this explicit for the reader (lines 108-112/144-148). Likewise, this reviewer objects to our focus on H2O2 rather than including other ROS forms; we have clarified the rationale for this editorial choice (lines 168-172/210-215).
This reviewer objects to some of the features of the INTRODUCTION, including some of its informality. We have extensively revised this section to address this criticism. These changes also eliminate some text that the reviewer construes as redundant. However, we emphasize that the interrelationships between the LMRS steps reviewed in sections 3.1-3.3 inevitably impose a small amount of cross-referencing in the text; though this cross-referencing may appear redundant at first glance, it is, nonetheless, essential for the clarity and coherence of the argument.
This reviewer wants more clarity about what we can anticipate about realistic clinical opportunities in view of the LMRS treatment resistant reviewed in this manuscript. We have revised lines 478-485/538-545 to make this more clear. We also created a Table (Table I) as requested by this reviewer, highlighting preclinical work of especially direct value to oncological clinicians making trial choices going forward. The punchline is that, collectively, the evidence indicates that LMRS targeting can plausibly be expected to enhance the clinical response to every major carcinoma treatment modality. Thus, future clinical options implied by current work are numerous and diverse. Note that the insights discussed herein are sufficiently new that no clinical trials directly exploiting them are yet underway. Stimulating such trials in the near-term future is among the most urgent motivations for this manuscript.
Reviewer 2 Report
The authors, Bingham. et al., aimed to explore the progress made in comprehending lipid metabolism inside advanced carcinomas, specifically focusing on its implications for cancer metabolism adaptation which leads to resistance to many therapeutic interventions. This objective involves using FDA-approved medications to address resistance to cancer treatment. It was addressed in their review article titled "Toward a unifying hypothesis for redesigned lipid catabolism as a clinical target for advanced, treatment-resistant carcinomas."
This constitutes a comprehensive, and interesting topic of review work that is appreciated. However, there are major concerns that can be resolved to improve the quality of the manuscript, listed below.
1. The introduction should provide a more comprehensive discussion on how your work is adding novel importance to the current body of research. The authors' conceptualization of adipocyte remodeling, the resistance system of lipid metabolism, and its clinical application seems to be intricate, resulting in potential perplexity among readers. The authors should certainly consider revisiting the throughout section.
2. There is an excessive amount of auxiliary writing included in each part, which is causing the core scientific material to become diluted. Please rewrite the portions (line: 26-64).
3. In between what is meant by [reviewed in 8-10], [see, for example, 11], [reviewed in 12-14], etc.
4. References are mentioned very confusing manner. Please rectify that.
5. Please Concise the sub-headings.
6. The entire manuscript is in its earliest stages. As a final draft, the finalization requires extensive revision.
There is room for grammatical improvement.
Author Response
Bingham and Zachar ijms-2562173
Responses to REVIEWER 2
This reviewer was unsatisfied that our manuscript oriented the reader adequately. He/she also objects to “auxiliary writing.” We have extensively revised the INTRODUCTION to address these issues. Moreover, we have also made a number of small revisions throughout the manuscript to improve the clarity and economy of the text - to reduce the reader “perplexity” to which this reviewer refers (see Track Changes).
This reviewer is concerned that our referencing is unclear in places. We have revised all cases where this might be true to make the manuscript more reader friendly.
Excessive lengths of our lowest order subtitles were mentioned by this reviewer. We presume this comment results from the fact that IJMS software combined the subtitles and the first paragraph of following text in to a “giant title” in places. We have revised the text with a view to preventing this software artifact in this draft.
This reviewer expresses a general concern about the clarity of the English in the manuscript. We have looked it over carefully and had colleagues give it a another read. We have revised anything we can find that might be confusing. We emphasize that both authors have elite English expertise, writing and speaking. We are both highly educated and have four decades of experience in advanced scientific writing. Moreover, one of us is a fully native American English speaker and the other converted to American English as sole daily language in childhood. Perhaps this reviewer is not a native American English speaker and is put off by some of our word/grammar choices. We would be willing to consider revising any specific additional sentences/paragraphs this reviewer wants to call our attention to.
Round 2
Reviewer 1 Report
In my opinion the review is suitable to be published in IJMS: the authors had improved it in its meaning.
Minor detail: in table the authors should postpone in vivo studies to the in vitro ones.
Author Response
Bingham and Zachar ijms-2562173
Responses to REVIEWER 1 – second cycle
Technical note: We accepted the last round of revisions so that new revisions made here are clearly visible in Track Changes
We are gratified that this reviewer finds our revised manuscript ready to publish.
We have revised the Table according to this reviewer’s request.
Reviewer 2 Report
The authors, Bingham. et al., aimed to explore the progress made in comprehending lipid metabolism inside advanced carcinomas, specifically focusing on its implications for cancer metabolism adaptation, which leading resistance to many therapeutic interventions. This objective involves the use of FDA-approved medications to address resistance to cancer treatment and was addressed in their review article titled "Toward a unifying hypothesis for redesigned lipid catabolism as a clinical target for advanced, treatment-resistant carcinomas."
This constitutes a comprehensive, and interesting topic of review work that is appreciated. However, there are still some concerns that should be resolved to improve the quality of the manuscript, listed below.
1. A more concise summary is still needed in the introduction.
2. Each section still has an excessive quantity of ancillary writing, which is leading the primary scientific substance to get watered down. Kindly rewrite the sections.
3. The reference concern is not yet addressed and is provided in a quite unclear way. Could you please correct that?
4. The whole work does not have a sound that is consistent with the scientific approach.
It is important to incorporate scientific perspectives and use appropriate scientific expression.
Author Response
Bingham and Zachar ijms-2562173
Responses to REVIEWER 2 – second cycle
Technical note: We accepted the last round of revisions so that new revisions made here are clearly visible in Track Changes
We edited the Introduction and other sections significantly in response to this reviewer. We attempted to address his/her concerns as well as we could, in view of the limited detail in the review.
As well, we have further simplified referencing, though with possible loss of useful supplementary information for the reader.
Round 3
Reviewer 2 Report
I would recommend the manuscript for publication. The authors addressed the section of the manuscript as per the raised concerns.